# Healthcare Service Utilisation Across Continuum of Care for Type 2 Diabetes Among Culturally and Linguistically Diverse Populations: A Systematic Review

**DOI:** 10.3390/ijerph22081279

**Published:** 2025-08-15

**Authors:** Mahfuzur Rahman, Resham B Khatri, Sjaan Gomersall, Mosharop Hossian, Asaduzzaman Khan

**Affiliations:** 1School of Health and Rehabilitation Sciences, The University of Queensland, Brisbane 4072, Australia; s.gomersall1@uq.edu.au (S.G.); m.hossian@uq.edu.au (M.H.); a.khan2@uq.edu.au (A.K.); 2School of Public Health, The University of Queensland, Brisbane 4006, Australia; r.khatri@uq.edu.au; 3Health and Wellbeing Centre for Research Innovation, School of Human Movement and Nutrition Sciences, The University of Queensland, Brisbane 4072, Australia

**Keywords:** culturally and linguistically diverse, type 2 diabetes, continuum of care, healthcare service utilisation

## Abstract

*Introduction:* Healthcare service utilisation for type 2 diabetes (T2D) is suboptimal among people from culturally and linguistically diverse (CALD) backgrounds. Synthesised evidence on drivers influencing their healthcare access and utilisation across the continuum of care (CoC) is scarce. This review synthesised drivers of accessing and utilising healthcare services across the CoC for T2D from the perspectives of people from CALD backgrounds and their healthcare service providers (HSPs). *Methods:* Five databases (Scopus, PubMed, Web of Science, CINAHL, and PsycINFO) were searched from inception to November 2023. This review was prospectively registered with PROSPERO (ID: CRD42023491560). The McGill Mixed Methods Appraisal Tool (MMAT) was used to assess the quality of articles included in this systematic review. Studies were included if they were primary research involving people from CALD backgrounds and/or their HSPs, and reported data related to access to and utilisation of services across the CoC. The CoC framework was used to conduct a guided thematic analysis for qualitative findings and narrative synthesis was used to summarise quantitative findings. *Results:* Thirty-five studies were included: 31 qualitative, 3 quantitative, and 1 mixed-method. Psychosocial (e.g., fear of treatment) and sociocultural barriers (e.g., stigma) were reportedly encountered at diagnosis and initiation of treatment. Language and cultural barriers were expressed by most people with T2D and their HSPs, especially at the monitoring and adherence stages of the CoC. Trusted relationships with HSPs and the cultural competency of the HSPs were facilitators of continued monitoring and adherence and long-term care and management. No barriers or facilitators were identified for decision-making to enter the healthcare systems, screening, and first point of contact aspects of the CoC from the perspectives of either patients or HSPs. *Conclusions:* Although included articles were heavily skewed towards qualitative studies, the synthesised evidence on factors influencing access to and utilisation of healthcare services for T2D can inform policymaking by highlighting pathways to improved use of healthcare services among these groups.

## 1. Introduction

Globally, the prevalence of diabetes mellitus is rising. According to a report of the International Diabetes Federation (2021), 540 million people worldwide have diabetes [1]. By 2050, it is estimated that more than 1.31 billion people will likely have diabetes, with type 2 diabetes (T2D) accounting for over 90% of all diabetes cases [2]. However, the prevalence of T2D is not evenly distributed; for example, there is higher prevalence in groups such as those with different cultural backgrounds and immigration status [3,4].

People from culturally and linguistically diverse (CALD) backgrounds include those born in and migrated from non-English speaking countries and where language other than English is the main language spoken at home [5,6]. However, in this review, we also included people as CALD who migrated from one non-English speaking country to another non-English speaking country and had distinct cultural backgrounds. T2D is prevalent in people with CALD backgrounds across Europe, the United States, and Australia [7,8,9]. A study conducted in the United States estimated that the prevalence of T2D was 22.1% for Hispanics and 19.2% for Asians, whereas it was 12.1% for non-Hispanic white [8]. Another study carried out in the United Kingdom showed that the prevalence of T2D among different CALD groups—such as Caribbean, African, Indian, Pakistani, and Chinese—was approximately 3–5 times higher when they lived in Britian than their white British counterparts [7]. According to the Australian Bureau of Statistics, in 2022, the prevalence of T2D in Australia was 6.7% among people from CALD backgrounds, whereas it was 4.7% among their counterparts with non-CALD backgrounds [9].

People with CALD backgrounds not only have a higher prevalence of T2D, but they are also more prone to experiencing T2D-related health consequences [10]. For example, migrants from Northeast Asia and Southeast Europe living in Australia had higher levels of psychological distress and other type 2 diabetes complications (e.g., poor eyesight) compared to people born in Australia [10]. A systematic review revealed that people with T2D from CALD backgrounds experienced comparatively poorer health outcomes due to a range of factors, including socioeconomic status, suboptimal lifestyle behaviours and self-management, and barriers to healthcare services compared to their non-CALD counterparts [11].

Barriers to accessing healthcare services among people with T2D from CALD backgrounds are likely to be multifaceted. One systematic review synthesised the evidence on psychosocial barriers (e.g., cultural beliefs) to accessing healthcare services for T2D [12]. However, this review did not indicate how the barriers could differ at different stages of the continuum of care (CoC). Further, although the review included studies on several ethnic minorities, it was not focused on people from CALD backgrounds [12]. Another narrative review provided evidence on the barriers to accessing healthcare services specifically among people from CALD backgrounds, reporting factors such as language and communication challenges and the ‘healthy migrant’ effect (i.e., migrants typically tend to have a better health than natives) [13]. However, this review was not focused on T2D.

Diabetes is a medical condition that requires a wide range of healthcare services for an individual across the CoC, which are delivered by a range of multidisciplinary service providers (e.g., general practitioners and accredited diabetes educators) at different stages for different healthcare services [14]. A systematic review demonstrated that receiving or practicing higher level of CoC for T2D was associated with improved health outcomes, such as a significant improvement in haemoglobin A1c levels and reduced hospitalisation rate [15].

The CoC is a framework that is being used to ensure a comprehensive approach to identifying gaps at each stage of seeking, accessing, and receiving consistent, appropriate, and quality care for non-communicable diseases like hypertension, cardiovascular disease, and diabetes (Figure 1) [16,17,18]. The Medtronic Foundation initially conceived this framework, and the Institute of Health Metrics and Evaluation (IHME), in collaboration with the Medtronic Foundation, evaluated a multicountry program named the HealthRise using this approach [19]. The framework has been used to evaluate access to chronic care in underprivileged communities in Brazil, India, South Africa, and the United States [19] and has been instrumental in capturing information about the barriers to diagnosis, treatment, and management of chronic diseases from the perspectives of healthcare receivers and providers [16,18].

While previous systematic reviews have synthesised evidence on barriers and facilitators of healthcare service utilisation among people from CALD backgrounds, none have explored their intersections or employed the CoC framework. This is the first systematic review to identify and synthesise both barriers and facilitators of healthcare utilisation for T2D among CALD populations across each stage of the CoC, incorporating perspectives from individuals with T2D and healthcare service providers (hereafter referred to as HSPs). By applying the CoC framework and integrating insights from both patients and providers, this review offers a holistic understanding of the factors influencing T2D management and healthcare access within CALD communities. Additionally, it highlights crucial gaps in the global literature at three key stages of the CoC, providing directions for future empirical studies.

## 2. Materials and Methods

### 2.1. Design and Framework

The reporting of this systematic review follows the Preferred Reporting Items for Systematic Reviews and Meta-Analyses (PRISMA) criteria [20] and was prospectively registered with PROSPERO (ID: CRD42023491560). Our review was guided by the CoC framework, which was initially conceived by the Medtronic Foundation, and the Institute of Health Metrics and Evaluation [17,19].

### 2.2. Search Strategy and Study Selection

Searches were conducted from their inception to November 2023 in five databases: Scopus, PubMed, Web of Science, CINAHL, and PsycINFO. Search terms were identified and organised under four key concepts: drivers (barriers and facilitators), continuum of care, type 2 diabetes, and culturally and linguistically diverse. Search terms, including Boolean operators, are as follows:

(Driver* OR “Communication Barriers” [Mesh] OR enabler* OR Barri-er* OR obstacle* OR challenge* OR difficult* OR facilitator* OR support* OR constraint* OR success* OR “linguistic barrier*” OR “cultural barrier*” OR “attitudinal barrier*” OR “environmental barrier*”) AND ( “culturally competent care” [Mesh] OR “Patient Acceptance of Health Care” [Mesh] OR “Health Services Accessibility” [Mesh] OR “Health Ser-vices” [Mesh] OR “Continuity of Patient Care” [Mesh] OR “physician-patient relations” [Mesh] OR “patient care team” [Mesh] OR “delivery of health care, integrated” [Mesh] OR “healthcare utili*” OR “health service utili*” OR “health* use” OR “health service use” OR “service use” OR “service utili*” OR prevention OR treatment OR diagnosis OR screening OR promotion OR “continuum of care” OR “continuity of care” OR “care continuum” OR “relational continuity” OR “longitudinal care” OR “coordinated care” OR “care continuity” OR “co-ordinated care” OR “care coordination” OR “care co-ordination” OR “transition of care”) AND (diabetes OR “diabetes mellitus” OR “type 2 diabetes” OR “type 2 diabetes mellitus” OR “Diabetes Mellitus” [Mesh] OR “Diabetes Mellitus, Type 2” [Mesh]) AND (CALD OR multicultural OR migrant* OR immigrant* OR refugee* OR ethnic* OR “non-English speak*” OR NESB OR “racial divers*” OR multi-cultural OR “Asylum seeker*” OR “minority groups” [Mesh] OR “ethnic and racial minorities” [Mesh] OR “cultural diversity” [Mesh])

The first author (MR) developed the search terms and strategy using appropriate Boolean operators and control vocabularies. All authors reviewed the search terms. Finally, a university research librarian verified the search terms and strategy.

Studies were eligible for inclusion in the review if they were (a) empirical research (quantitative, qualitative, or mixed-method) published in English, (b) published in peer-reviewed journals, (c) included study participants from CALD backgrounds including immigrants, ethnic groups, and refugees with type 2 diabetes, and (d) and (e) reported drivers (facilitators and barriers.) for access to and utilisation of T2D services across the CoC. Studies were excluded if they (1) reported type 1 or gestational diabetes, or (2) were grey literature (e.g., commentaries/theses), letters to the editor, book chapters, opinions, or editorials. Studies were imported into the online tool Covidence (Veritas Health Innovation Ltd., Melbourne, Australia), which was used to conduct the screening process. Title and abstracts of the returned studies were independently assessed for eligibility compared to the inclusion and exclusion criteria by two of the authors (MR and MH). If the papers were considered eligible, the full text was then evaluated. Any disagreements were settled by discussion. In case of dispute, the issue was discussed with other co-authors until agreement was reached (AK, SG and RBK).

### 2.3. Quality Appraisal

The McGill Mixed Methods Appraisal Tool (MMAT) was used to assess the quality of articles included in this systematic review [21]. The MMAT version 18 comprises five domains, and there are five questions under each domain Appendix A. If the response to a question was ‘Yes’, it was coded as 1; if the response to a question was ‘No’, it was coded as 0; if the response to a question was partially provided, it was coded as ‘0.5’; and if the response was ‘cannot tell’, it was coded as ‘CT’. Hence, the total score varies from 1 to 5. Two reviewers (MR and MH) assessed the quality of the included papers. Disparities were resolved through discussion between the two reviewers and the other co-authors.

### 2.4. Data Extraction and Synthesis

Data were extracted by the first author (MR) using a matrix in Microsoft Excel software. The matrix organised extracted data into barriers or facilitators reported in the different stages of the CoC (decision to enter the health systems, first point of contact, screening, initiation of treatment, continued monitoring and adherence, long term management and disease control). It also included information on the author(s), year of publication, country, journal name, study participants, study design, and sample size. If any papers findings were unclear or missing, MR contacted the original authors for clarity or to minimise unreported data. Lastly, a sample (20%) of data were cross-checked by one of the authors (MH) to verify accuracy of the data extraction.

Qualitative findings were analysed using both an inductive and deductive thematic approach. At first, data were synthesised under a priori themes: barriers to access and utilisation, and facilitators of access and utilisation of T2D services. Then, emerging themes and sub-themes were identified inductively. The emerging themes were socio-cultural, psychological, financial, legal, and infrastructural or facility-based.

Quantitative findings were analysed using the subset of the included studies that reported descriptive statistics relevant to study objective. Narrative synthesis of the quantitative data was conducted and triangulated with qualitative data as and when appropriate.

## 3. Results

The study PRISMA flowchart is presented in Figure 2. Thirty-five studies were included: 31 were qualitative; three were quantitative; and one was a mixed-methods study.

Among the included articles, 26 were from the perspectives of people with T2D from CALD backgrounds, four were of their HSPs, and five were of both people with T2D from CALD backgrounds and their HSPs. HSPs included individuals providing services for T2D (e.g., general practitioners, pharmacists, dieticians, diabetes educators, health nurses, podiatrists). The characteristics of the included studies are described in Table 1.

### 3.1. Barriers and Facilitators

Quantitative and qualitative findings related to the barriers and facilitators of T2D along the CoC from the perspectives of T2D patients from CALD backgrounds and their HSPs were synthesised into themes, following the stages of the CoC (Table 2 and Table 3).

The synthesised evidence focuses largely on the barriers and facilitators across four stages of the CoC: (i) diagnosis, (ii) initiation of treatment, (iii) monitoring and adherence, and (iv) long-term care and treatment among people from CALD backgrounds.

### 3.2. Diagnosis

Thematic analysis identified three main barriers, and no facilitators, to diagnosis from the perspective of people with T2D from CALD backgrounds. Sociocultural barriers to diagnosis or reasons for delayed diagnosis reported in the literature from the perspective of patients from CALD backgrounds included social stigma [23,45,47] and religious misperceptions [23]. Studies reported that due to social stigma (e.g., people gave a different look to a T2D patient after hearing that someone had T2D), patients from CALD backgrounds did not want to expose their disease status to others, which in turn resulted in delayed diagnosis for T2D [23,45,47].

Religious misperceptions (e.g., T2D as a condition, a sign of being close to God) [23] also delayed diagnosis of T2D among people from CALD backgrounds. Psychological barriers such as fear of tests and procedures, and fear of treatment due to unfamiliarity with Western healthcare systems also delayed diagnosis for T2D among patients from CALD backgrounds [52]. Inadequate knowledge and misperceptions about the symptoms of T2D (e.g., perception that symptoms were due to depression) [23,25,53,55] were also reported as barriers to diagnosis for T2D among patients from CALD backgrounds. Financial constraints [55], such as lack of insurance coverage, also resulted in delayed diagnosis for T2D among patients from CALD backgrounds. Financial constraints are particularly prevalent among disadvantaged CALD populations (e.g., among rural African Americans in the United States) [55].


*“Diabetes is a big stigma-related problem in our community, especially among the Black African community, because of the lack of awareness and ignorance of people. Going for treatment or diagnosis at the hospital is not always the first call for people because of the stigma. People self-diagnose, self-medicate, and some ignore it because of beliefs or other family commitment”*

*(An African participant living with T2D in the UK [47])*


None of the included studies reported barriers or facilitators to diagnosis or delayed diagnosis of T2D among patients from CALD backgrounds from the perspectives of HSPs.

### 3.3. Initiation of Treatment

Sociocultural, psychosocial, and infrastructural issues emerged as the barriers to initiation of treatment for T2D among people with CALD backgrounds [33,47,48]. Initiation of treatment for T2D among patients from CALD backgrounds was impeded by social stigma and fear of treatment for T2D [33,47]. A lack of awareness about the severity of T2D due to its non-visible signs resulted in patients with T2D from CALD backgrounds being reluctant to initiate treatment [33]. Distrust of Western medicine and easier access to traditional medicine also came up as barriers to treatment among T2D patients from CALD backgrounds [33,48].

In some cases, patients with T2D from CALD backgrounds reported that they did not initiate treatment due to a lack of information and perceived harm of treatment [56]. For example, there is a lack of information about the treatment for diabetic retinopathy and their perceptions that even treatment for diabetic retinopathy might cause harm to their eyesight [56].


*“It’s a question of ethics, but also professionalism because if they make a mistake in the manipulation of the products, it can maybe cause a complete loss of eyesight.”*

*(A Pakistani T2D patient living in Canada [56])*


Apart from the socio-economic status of the family, family members’ perceived risk of T2D could also positively influence and facilitate the initiation of treatment, as is evident among Sub-Saharan and African immigrants living in the UK [47].

Psychological issues [52] and insufficient understanding about the severe consequences of T2D [49] were expressed as barriers to initiating treatment from the perspectives of HSPs.


*“I feel like they’re still a little bit hesitant about the western medication, which can be, you know, scary at times for them”*

*(A Hmong American case manager providing services in the USA [48])*


HSPs also reported barriers to initiating treatment for T2D among refugees, and these were mostly pertaining to infrastructures and facilities [41,43]. These barriers included transportation costs, housing instability [43], and legal issues [41].


*“A patient whose immigration case was in the process of being reviewed, and who was afraid that revealing her chronic illness by requesting government assistance to obtain supplies would jeopardise her chances of gaining US citizenship.”*

*(A Diabetic Educator in the USA [41])*


Socioeconomic status and positive influence by the family members came up as the facilitators of initiation of treatment for T2D among people with CALD backgrounds [43,47]. However, synthesised findings lacked evidence on the facilitators of initiation of treatment for T2D from the perspectives of HSPs.

### 3.4. Continued Monitoring and Adherence

Language and cultural barriers emerged as the major barriers to continued monitoring and adherence to treatment for T2D among patients from CALD backgrounds followed by perceived insufficient empathy from GPs [34] and fear of side-effects of medication [25]. Due to language barriers, T2D patients with CALD backgrounds felt discouraged from visiting their GPs because they could not express their problems [23]. Particularly, it magnified the barrier in the absence of professional interpreters during consultation [30]. Cultural insensitivity in the advice given by GPs led to nonadherence to seeking treatment from GPs among people from CALD backgrounds [30,45].


*“The doctor doesn’t know what Karela (South Asian vegan recipe) is, you see, so he must give his advice grounded on his culture, his food-tradition, not ours”*

*(A Pakistani T2D patient living in Norway [30])*


Due to language barriers, GPs sometimes did not understand patients’ concerns about diets or alternative food choices [27]. As a result, the patients were not satisfied with their GPs’ recommendations and decided not to follow them [27].

Perceived insufficient empathy from GPs and their lack of attention to patients, compared to their home country, made the T2D patients with CALD backgrounds feel discouraged from further visits [34].


*“I don’t have money to buy the medicine, nor do I have a doctor to prescribe it. Here [referring to the free community clinic], we [referring to Hispanics] used to get medical attention, but not anymore.”*

*(A Hispanic woman living in the USA [34])*


Patients with T2D from CALD backgrounds also faced challenges in follow-up appointments with their GPs due to dissatisfaction with being examined by different GPs at each follow-up visit and getting inconsistent information from each one [25], women’s dependency on their spouse [53] and transportation issues, particularly for women [53].


*“…Women did not use buses as they could not ask for directions or understand when to get off; moreover, buses were considered unsafe for women for fear of racist abuse or vulnerability to mugging”*

*(Bangladeshi women with T2D, living in the UK [53])*


They also missed appointments due to the cost and accessibility to the facilities [56].

Nonadherence to treatment for T2D also stemmed from fear of side effects of medication [25] and demotivation by information provided by their peers [24].


*“I don’t go to that doctor anymore because I heard the medication, they gave my friend stopped his kidneys working.”*

*(A Maltese patient with T2D living in Australia [24])*


Language and cultural barriers to continued monitoring and adherence were also echoed from the perspectives of the HSPs, although their expressions differed [22,36,48]. HSPs expressed that patients could not understand what they explained [48] and that the communication gap (e.g., word or phrase used by HSP that was understood wrongly by the patients) created challenges among the patients and the HSPs [22,36]. Dieticians also faced difficulties if they perceived that they were not culturally competent [36].


*“You ask questions but don’t know enough about particular habits to adequately respond to that. With Dutch people, often you know what they do and what they don’t do, but with migrants, this is more difficult”*

*(A Dutch dietician [36])*


Facilitators were synthesised from the perspectives of both patients and HSPs. Patients expressed that the cultural competency of the HSPs [23,45], acculturation and cultural integration (e.g., being more familiar with healthcare services with cultural support in host countries) of T2D patients from CALD background [24], and reminders by the HSPs about follow-up [44] could facilitate their continued monitoring and adherence to treatment for T2D.


*“The first dietitian that I saw said, ‘Oh don’t take this, don’t take that’… then I was so anxious. [It] makes me not want to listen. But the second dietitian I spoke to know about our food. She had a model of fufu, so it makes me more welcome… that makes me more compliant listening to her because she knows what we eat.”*

*(An African woman with T2D living in the UK [45])*


Role of family members [56] and community health workers [32] also influenced patients engaging with continued monitoring and adherence to their treatments for T2D.

It is evident from the perspectives of both the patients [23,51] and HSPs [36] that trusted relationships between HSPs (e.g., GPs, dieticians) and T2D patients from CALD backgrounds could influence patients’ continued monitoring and adherence to treatment for T2D.


*“…because it’s really important that all my records and things are kept with the one doctor that I know”*

*(A Tongan woman with T2D, living in Australia [51])*


### 3.5. Long-Term Care and Management

Language and culture were also identified as prominent barriers to long-term care and management of T2D among people with CALD backgrounds [23,37,40,41,46], followed by patient–provider gender discordance [26,53] and transportation and financial problems [27,37,40].

Patients with T2D from CALD backgrounds reported language and culture as barriers to accessing information relevant for long-term care and management of their conditions [23,37,40,41,46]. Due to language barriers, they had a lack of access to information, and thus, they did not know if there were diabetes education materials, diabetes support groups, and education sessions available for them [23,40,41,46]. It was also difficult for them to express their disease progress to the HSPs [37,46,52]. Since the patients with T2D from CALD backgrounds could not explain their problems and understand advice from their HSPs, patients then did what they thought was best [29,54].


*“When it isn’t explained properly, you just do what you think is best… people need to be given advice that is closer to their own culture and language.” *

*(A Turkish patient with T2D, living in Norway [29])*


In addition, patient–provider gender discordance came up as a barrier to seeking long-term care and treatment for the patients, particularly women with T2D from CALD backgrounds [26,53]. Although most of the women with T2D were found to be happy with the services they were receiving (regular checks for retina or foot), they expressed their concerns in this regard [53].


*“No, the doctor (male) always rushes. I tell my daughter-in-law all my problems then she tells the nurse (female) everything openly. It’s easier to tell another woman but with the doctor it’s not possible”*

*(A Bangladeshi woman with T2D, living in the UK [53])*


Patients with T2D from CALD backgrounds did not know where to visit for the long-term management of their conditions, because their GPs did not refer them or inform them of the availability of additional T2D services [24,29,34,44,50,55]. There was also reportedly limited access to community resources such as diabetic education for T2D patients from CALD backgrounds, as they expressed [37].


*“I didn’t know I needed to get my feet checked. I normally check my feet at home, the doctor doesn’t have time to teach you all the things you need to know” *

*(A Maltese patient with T2D, living in Australia [24])*


Transportation and financial problems also created difficulties in engaging with long-term care and management of T2D among CALD populations [27,37,40]. Financial problems, difficulties in finding same-sex HSPs, and lack of information on where to go, were barriers that were more prominent among women from CALD backgrounds compared to their male counterparts [35].

Psychological issues such as patients perceived authoritative behaviour of the HSPs also came up as a barrier to long-term care and treatment for them [51,54], with patients with T2D from CALD background becoming frustrated that they were passive receivers of treatment [54].

Language barriers were echoed from the perspectives of the HSPs [28,49]. The HSPs faced difficulties in providing advice on managing T2D among patients from CALD backgrounds [28,49]. They also expressed some barriers, such as a lack of training for managing T2D patients from CALD backgrounds for their long-term care [49].

The synthesised findings lacked the evidence of facilitators of long-term care and management for T2D among people with CALD backgrounds, from the perspectives of both the patients and HSPs.

### 3.6. Narrative Synthesis of Quantitative Findings

Across three quantitative and one mixed-methods studies [35,39,42,43], drivers (facilitators and barriers) regarding continued monitoring and adherence [39] and of long-term care and management for type 2 diabetes among people from CALD backgrounds [35,39,42,43] were identified. Socioeconomic disadvantage, particularly household food insecurity, was associated with barriers to accessing doctors such as transportation difficulties (OR 1.22; 95% CI 1.04–1.43) [39]. Psychological distress, indexed by higher depression scores, was associated with higher odds of forgetting appointments (OR 1.04; 95% CI 1.01–1.06) [39]. Social resources operated as facilitators: support from family and friends, self-efficacy, and the ability to cover health insurance improved access; family and friend support specifically reduced the odds of barriers to visiting a doctor (OR 0.93; 95% CI 0.87–0.99) [39]. Among refugees, housing instability was associated with lower odds of seeking GP care (OR 0.48; 95% CI 0.23–1.00) [43], and higher monthly expenditures were associated with greater odds of seeking specialist care (OR 1.46; 95% CI 1.10–1.91) [43]. Vulnerable subgroups faced more barriers, with women reporting more barriers than men (1.9% vs. 1.2%; *p* = 0.06) [42].

## 4. Discussion

This systematic review of the literature identified a wide range of barriers to and facilitators of access to healthcare service utilisation across the CoC for T2D among people from CALD backgrounds. Barriers and facilitators were identified for diagnosis, initiation of treatment for T2D, continuing monitoring and adherence, and long-term care and management. Identified barriers and facilitators were from the sociocultural, financial, legal, and infrastructural perspectives of the CoC and were varied in terms of gender and socioeconomic status within the patients from CALD backgrounds. No barriers and facilitators were identified for decision-making to enter the healthcare systems, screening, and first point of contact aspects of the CoC from the perspectives of either patients or HSPs. In this review, evidence related to the drivers for access to and utilisation of healthcare services for T2D among people from CALD backgrounds was predominantly from the perspectives of people with T2D, rather than their HSPs. Further, while the perspectives of some HSPs were captured (e.g., general practitioners, dieticians, nurses), these did not encompass the viewpoints of all potential HSPs involved in T2D care services.

Among the barriers to accessing and utilising care for T2D patients from CALD backgrounds, synthesised in this review, the most prominent across the CoC were understanding and communication in English and perceived cultural barriers. This finding corresponds with the findings from a systematic review that separately reported barriers to access to healthcare service utilisation for T2D in CALD populations [12]. However, that review did not explore the drivers of access to and utilisation of healthcare services across the CoC for T2D. Our review demonstrated that misunderstanding and lack of communication due to language and cultural barriers are not uniformly present at all stages of the CoC for T2D patients from CALD backgrounds. These barriers predominantly existed at continued monitoring and adherence, as well as at long-term care and management stages of the CoC for T2D patients from CALD backgrounds. In comparison, the more prevalent barriers to the initial phases of the CoC, such as diagnosis and initiation of treatment for T2D patients from CALD backgrounds, mostly stemmed from individual factors, such as psychosocial barriers. These include experiencing stigma, fear, distrust in western medicine or treatment, and unawareness of the disease. One study has revealed that stigma should not be analysed as an individual factor but rather as a socio-cultural aspect involving social acceptance and interactions between stigmatised and non-stigmatised individuals [57].

Barriers to access and utilisation of healthcare services were distinct among different groups within CALD populations [58,59], emphasising the importance of not viewing this as a homogenous group. Our review found that among T2D patients from CALD backgrounds, refugees, and women were more vulnerable and more likely to experience barriers. Refugees’ barriers along the CoC were mostly pertaining to their legal issues and housing instability. Women’s barriers were related to socio-cultural norms, gender roles, and resultant dependency on other members of their families. Using an intersectionality framework to better explore the role of multiple social positions of people with T2D from CALD backgrounds could be pivotal in understanding the drivers of healthcare service utilisation among them.

A unique contribution of the current study is simultaneously including the perspectives of patients and HSPs. Findings from this review suggest that HSPs emphasised language and cultural barriers influencing their ability to provide care for T2D patients from CALD backgrounds. Barriers expressed by the HSPs were based on their interactions with their service receivers. These barriers correspond with the findings of other studies, although those studies focus on continued monitoring and adherence to the CoC for T2D in the broader population rather than addressing people from CALD backgrounds [60,61]. As discussed earlier, the articles included in this review did not encompass the viewpoints of all potential HSPs who are likely to be part of a multidisciplinary type 2 diabetes care team. Therefore, further research is warranted to investigate the perspectives of other HSPs (e.g., endocrinologists, ophthalmologists, exercise physiologists, physiotherapists, psychologists, and social workers/counsellors) whose opinions were not covered in this review.

Among the studies in this review, few reported facilitators of access and utilisation of T2D care among people from CALD backgrounds. From the perspectives of the HSPs, a trusted relationship between patients and HSPs could foster long-term care and management for T2D patients from CALD backgrounds, although a study added that it could also help patients in medication adherence for T2D patients in general [60]. However, both T2D patients from CALD backgrounds and their HSPs emphasised the need for culturally appropriate treatment, suggesting that cultural competence training for HSPs is important, which is supported by other literature [59]. Services that are culturally tailored and target specifically people from CALD backgrounds are more effective than mainstream services in a multicultural country [62]. This systematic review further demonstrated that services should not only be population-specific but also consider the drivers at different stages of the care for chronic diseases like T2D among people from CALD backgrounds.

This review revealed that research gaps existed relating to the earliest stages of the CoC (e.g., decision-making to enter the healthcare systems, screening, and first point of contact). These gaps might have resulted from a lack of engagement of people from CALD backgrounds in the studies and not properly addressing their needs while designing the studies [63].


*Implications for Policy, Practice, and Future Research*


Findings should be interpreted in the context of high-income health systems, particularly Australia, the United Kingdom, and the United States, where many people with T2D from CALD backgrounds live. In Australia and the United Kingdom, universal coverage tends to reduce direct financial barriers; however, entry and continuity may be constrained by primary care gatekeeping, registration requirements, variable availability of professional interpreters, limited cultural competence, and out-of-pocket costs for medicines, transport, and ancillary care. In the United States, insurance-based coverage, cost sharing, and immigration-linked eligibility rules may create financial and continuity barriers, although safety-net services, community health worker programmes, and charity care can facilitate access where available. Across all three systems, access is most consistently improved when funded interpreter services are guaranteed, culturally competent care is embedded, referral pathways are streamlined, care coordination and navigation are resourced, and transport and medication subsidies are provided. These system features should be considered when generalisation to other settings is attempted. For priority setting and resource allocation, available evidence should be used systematically [64]. Barriers and facilitators most prevalent at specific stages of the continuum of care (CoC) should be prioritised for cost-effective action, and practitioners working at each stage should be trained to address the dominant barriers and leverage the relevant facilitators.

Further empirical studies, both qualitative and quantitative, are warranted to examine drivers across all CoC stages, with emphasis on underexplored stages. Facilitators should receive greater attention, as they were underrepresented relative to barriers. Studies should report health-system context and migration status to support interpretation and generalisation.


*Study Limitations*


Although this review synthesised a wide range of barriers to and facilitators along the CoC for T2D patients from CALD backgrounds, some limitations should be considered. People from CALD backgrounds are heterogeneous (e.g., in terms of ethnicity, socioeconomic status). Therefore, our review may overlook particular studies of certain subgroups within these people from CALD backgrounds. Only studies published in English were included in this review, so it was likely to miss some relevant articles published in other languages. In addition, included articles were heavily skewed towards qualitative studies.

## 5. Conclusions

This review identified a wide range of sociocultural, financial, legal, and structural barriers and facilitators from the perspectives of people with T2D from CALD backgrounds and their HSPs across the CoC. It highlights the need of further research into the drivers of access to and utilisation of healthcare services across the CoC, particularly at the decision-making stages to enter the healthcare systems, screening, and first point of contact, where evidence is currently limited in the global literature. Policy and service delivery should fund interpreter services, reduce out-of-pocket costs, and improve resource navigation and care coordination at screening and first contact. Future studies should prioritise facilitator-focused, context-specific interventions at these stages and report health-system context and migration status.

## Figures and Tables

**Figure 1 ijerph-22-01279-f001:**
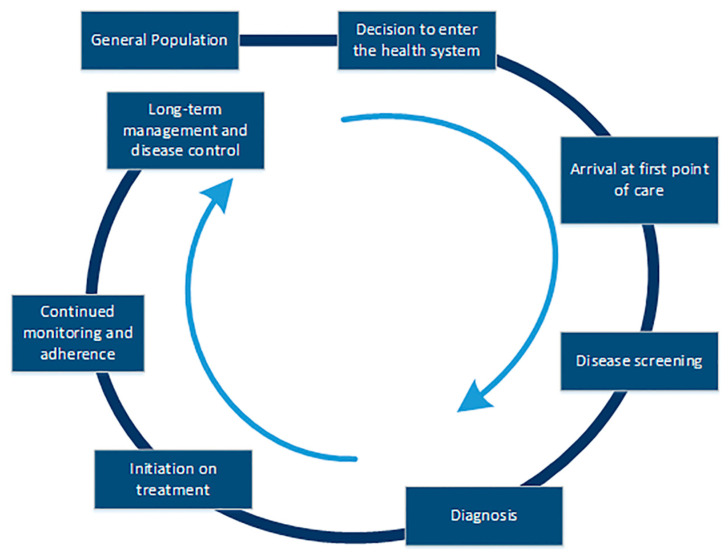
Continuum of care framework adapted from Gabert et al. 2017 [16] and Wollum et al. 2018 [18] (based on HealthRise program evaluation by the IHME).

**Figure 2 ijerph-22-01279-f002:**
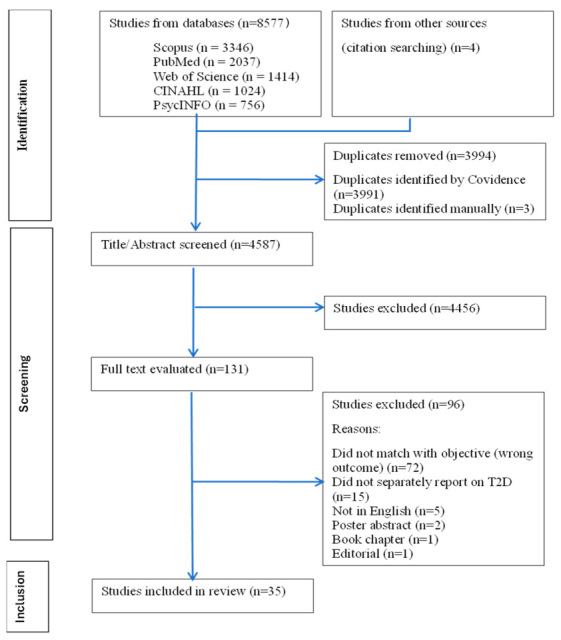
PRISMA flowchart.

**Table 1 ijerph-22-01279-t001:** Characteristics of included studies (arranged alphabetically by authors’ names).

Studies	Approaches	Settings	Perspectives (Patients/Caregivers or Healthcare Providers or Both)	Sample Size	Type of Participants
Almansour et al., 2017 [22]	Qualitative	Australia	Healthcare Providers	21	Community pharmacist
Alzubaidi et al., 2015 [23]	Qualitative	Australia	Patients	60	Arabs
Barbara et al., 2013 [24]	Qualitative	Australia	Patients	24	Maltese
Biyikli et al., 2017 [25]	Qualitative	Austria	Patients	13	Turkish
Carolan-Olah et al., 2018 [26]	Qualitative	Australia	Patients	13	Italians
Cha et al., 2012 [27]	Qualitative	USA	Patients	20	Korean Americans
Choi et al., 2018 [28]	Qualitative	Australia	Both	95 (Patients) + 15 (Healthcare Providers)	Chinese patients and dieticians
Cokluk et al., 2023 [29]	Qualitative	Norway	Patients	13	Turkish
Fagerli et al., 2005 [30]	Qualitative	Norway	Patients	15	Pakistani patients
Gele et al., 2015 [31]	Qualitative	Norway	Patients	30	Somalis
Heisler et al., 2009 [32]	Qualitative	USA	Patients	40	African American and Latinos
Ho et al., 2006 [33]	Qualitative	Canada	Patients	5	Chinese
Hu et al., 2013 [34]	Qualitative	USA	Patients	36 (Patients) + 37 (Caregivers)	Mexican Hispanics
Hyman et al., 2012 [35]	Quantitative	Canada	Patients	130	Asian migrants
Jager et al., 2020 [36]	Qualitative	The Netherlands	Healthcare Providers	12	Dutch, Bulgarian, and Turkish dieticians
Joo et al., 2016 [37]	Qualitative	USA	Patients	23	Korean Americans
Kokanovic et al., 2006 [38]	Qualitative	Australia	Patients	16	Greek, Indian, Chinese, and Pacific Island backgrounds
Kollannoor-Samuel et al., 2012 [39]	Quantitative	USA	Patients	211	Puerto Ricans and Latinos
Leung et al., 2014 [40]	Qualitative	USA	Patients	29	Chinese
Lipton et al., 1998 [41]	Qualitative	USA	Healthcare Providers	24	Mexican, Puerto Rican, and Hispanic healthcare professionals (GPs and nurses)
Lu et al., 2016 [42]	Quantitative	USA	Both	101 (Patients) + 44 (Healthcare Providers)	Hispanic and African American, healthcare providers and staff (not specified)
Lyles et al., 2022 [43]	Mixed methods	Lebanon	Both	373 (Refugees) + 24 (Healthcare Providers)	Syrian refugees, healthcare providers (GP)
Carolan-Olah et al., 2013 [44]	Qualitative	Australia	Patients	15	Vietnamese
Moore et al., 2022 [45]	Qualitative	UK	Patients	41	African, Caribbean backgrounds
Nam et al., 2013 [46]	Qualitative	USA	Patients	23	Korean Americans
Omodara et al., 2022 [47]	Qualitative	UK	Patients	36	Sub-Saharans and Africans
Park et al., 2023 [48]	Qualitative	USA	Both	13 (Caregivers) + 10 (Healthcare Providers)	Asian American, Hmong American caregivers and healthcare providers (not specified)
Patel et al., 2023 [49]	Qualitative	UK	Healthcare Providers	14	Asian healthcare providers (GPs, nurses, dieticians, podiatrists)
Ramal et al., 2012 [50]	Qualitative	USA	Patients	27	Hispanics
Kokanovic et al., 2007 [51]	Qualitative	Australia	Patients	30	Chinese, Indian, and South and Pacific Island backgrounds
Renfrew et al., 2013 [52]	Qualitative	USA	Both	15 (Patients) + 30 (Healthcare Providers)	Cambodian patients and healthcare providers (GPs, nurses, dieticians)
Rhodes et al., 2003 [53]	Qualitative	UK	Patients	12	Bengalis
Rose et al., 2015 [54]	Qualitative	Australia	Patients	28	Arabs and Vietnamese
Utz et al., 2006 [55]	Qualitative	USA	Patients	74	African Americans
van Allen et al., 2021 [56]	Qualitative	Canada	Patients	39	Pakistanis, Chinese, and Africans

**Table 2 ijerph-22-01279-t002:** Barriers to and facilitators of healthcare service utilisation for T2D from the perspectives of people with CALD backgrounds under different themes, following the stages of continuum of care.

Stage of Continuum of Care	Barriers	Facilitators
**Decision to enter the health system**	No data available	No data available
**First point of contact**	No data available	No data available
**Screening**	No data available	No data available
**Diagnosis**	Delayed diagnosis due to social stigma [23,45,47], fear of Western healthcare systems [52], misperceptions of seriousness of disease/symptoms [45,55], and lack of health insurance [55].	No data available
**Initiation of treatment**	Social stigma [33,47] and fear [33] of treatment. Perceiving T2D as not being a severe illness at initial stage since there are no obvious symptoms [33]. Easier access to traditional medicine [33]. Legal barriers (e.g., afraid of revealing T2D status when citizenship is in the process of being reviewed) [41]. Housing instability and transportation costs, especially for refugees [43]. Not understanding the ability or importance of treating complications, and perceived harm of treatment [56].	Perceived risk of T2D among family members [47] Subsidised treatment cost and early diagnosis (particularly for refugees) [43]
**Continued monitoring and adherence**	Due to language and cultural barriers, patients could not express their problems and thus feel discouraged from visiting HSPs [23]. Due to cultural insensitivity, patients did not follow their prescribed course of care, perceiving it to be irrelevant [30,45] Less trust in HSP made the patients not visit the HSP further [27] Perceived less attention or less sympathy by the HSP in host countries compared to experiences with HSP in home country [34] Difficulty attending follow-up appointments, particularly for women due to dependency on others and transportation issues [53] Missing appointments due to cost [56] Demotivated by peers’ information on the negative consequences of treatment [24] Fear of side-effects of the treatment [25]	Presence of a translator in the healthcare setting [24] HSP from the same cultural and language backgrounds [23] Trusted relationship with HSP encouraged the patients to visit the HSP further and share problems [23,51] Accessibility to HSP in terms of distance from residence to healthcare setting [24] Presence of and motivation from community health workers [32] Reminder of follow-up from HSP [44] Positively influenced by family members to attend follow-up visit [56]
**Long-term care and management**	Patients could express disease progress [37,46,52] Patients were unaware of the availability of education materials, diabetic support, education support, etc., due to a lack of information in their language [23,40,41,46] Lack of information about the required HSPs (nutritionists, podiatrists, diabetic education, specialists in rural areas, etc.) [24,29,34,44,50,55] Cultural insensitivity in GPs’ [29,54] and diabetes educators’ [48] approach (e.g., lack of culturally specific diabetes self-management information from GPs and dieticians) Perception that the GP is the main source of receiving treatment and so less likely to visit to other HSPs despite having need [24] Not referred by the GP to other HSPs (e.g., nutritionist, podiatrist) [24] Females were unable to use public transport alone and forego long-term care and management from HSP [40] Perceived authoritative behaviour of GP made the patient frustrated as a passive receiver and cannot express their own experiences [40,51,54] Due to physiological problems (visual or hearing problems), older patients could not access information and felt embarrassed to seek help for others [40]	General practitioner (GP) (general medical practitioner who is often the first point of contact) from own language and cultural group [43]

**Table 3 ijerph-22-01279-t003:** Barriers to and facilitators of healthcare service utilisation for T2D from the perspectives of healthcare providers under different themes, following the stages of continuum of care.

Stage of Continuum of Care	Themes-Barriers	Themes-Facilitators
**Decision to enter the health system**	No data available	No data available
**First point of contact**	No data available	No data available
**Screening**	No data available	No data available
**Diagnosis**	No data available	No data available
**Initiation of treatment**	Difficulty with ascertaining patient’s history (refugee case) [52] Perception that patients fear or have less trust in Western medication [48]	No data available
**Continued monitoring and adherence**	HSP perceives that patient cannot understand what the dietitian explains and so does not adhere to treatment [22,36] Dieticians are not familiar with patients’ food habits (cultural incompetency) and so it is difficult to monitor food habits and give advice [36]	Patients’ trusted relationships with dieticians encouraged the patients to attend follow-up visits [36]
**Long-term care and management**	Language and cultural barriers [49] Difficulties in registration and payment process due to language barrier [49] Lack of training in providing healthcare for ethnic communities [49] Individual diabetes education sessions are not attractive for patients and so they feel discouraged about long-term management [28]	No data available

## Data Availability

The data will be made available on reasonable request to the corresponding author.

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
