# Peer review of "Healthcare Service Utilisation Across Continuum of Care for Type 2 Diabetes Among Culturally and Linguistically Diverse Populations: A Systematic Review"

_ijerph, 2025, doi:10.3390/ijerph22081279_

Round 1
Reviewer 1 Report
Comments and Suggestions for Authors
Dear authors, thank you for the opportunity to review this paper examining the use of health services for type 2 diabetes among culturally and linguistically diverse populations across the continuum of care. I suggest major revision after which this manuscript would be considered for publication.
However, to increase the clarity, analytical depth, and practical utility of the paper, I kindly recommend the following revisions:
- Decision gaps in the early stages of the continuum of care. The review reveals a complete lack of findings for the earliest stages (entry decision, first point of contact, and screening). While this is acknowledged, the manuscript would have benefited from a more detailed discussion of why this gap exists. I encourage the authors to elaborate these gaps in the discussion and suggest how future research might address them.
- Improve integration of quantitative findings. While some quantitative studies are included, their synthesis is limited. Consider creating a separate subsection to better connect and interpret how quantitative data supports or diverges from qualitative themes.
- Consider visualizing the positioning of barriers and facilitators on a CoC framework map. Just as possibilities.
- As most of the included studies are from Australia, the UK and the US, it would be useful to consider how national healthcare systems and policy contexts may affect access barriers and facilitators. This is particularly important for generalization to global CALD populations.
- Develop the role of the facilitator. Facilitators are underrepresented compared to barriers and deserve further analysis.
- Strengthen the conclusion The conclusion could be more action oriented. Consider summarizing key recommendations for policy, service delivery, and future research. This will be very helpful for policymakers.
- Some parts of the Discussion and Results repeat earlier information. Consider tightening them up to improve clarity and focus. This will allow more room for the discussion I recommended earlier.
- Have you used any AI tool to improve the grammar and style of the text. I also suggest a careful review of the language, as some phrasing appears overly like source texts, possibly triggering plagiarism detection. Consider paraphrasing for clarity and originality.
- Try to emphasize the results more in the abstract.
Sincerely,
A.
Author Response
Comments from Reviewer 1:
Comment:
Dear authors, thank you for the opportunity to review this paper examining the use of health services for type 2 diabetes among culturally and linguistically diverse populations across the continuum of care. I suggest major revision after which this manuscript would be considered for publication.
However, to increase the clarity, analytical depth, and practical utility of the paper, I kindly recommend the following revisions:
- Decision gaps in the early stages of the continuum of care. The review reveals a complete lack of findings for the earliest stages (entry decision, first point of contact, and screening). While this is acknowledged, the manuscript would have benefited from a more detailed discussion of why this gap exists. I encourage the authors to elaborate these gaps in the discussion and suggest how future research might address them.
Response: We appreciate your observation regarding the research gaps in the early stages of the continuum of care. In the revised manuscript, we have expanded the discussion with the following texts with references:
“This review revealed that research gaps existed relating to the earliest stages of the CoC (e.g., decision making to enter the healthcare systems, screening and first point of contact). These gaps might have resulted from a lack of engagement of people from CALD backgrounds in the studies and not properly addressing their needs while designing the studies.’ (page17, paragraph-1, lines: 419-422)
Comment:
- Improve integration of quantitative findings. While some quantitative studies are included, their synthesis is limited. Consider creating a separate subsection to better connect and interpret how quantitative data supports or diverges from qualitative themes.
Response: We added a standalone Results subsection, “Narrative synthesis of quantitative findings,” that explicitly maps the principal findings from the three quantitative and one mixed-methods studies to the corresponding qualitative themes (e.g., socioeconomic hardship and access barriers, psychological distress and adherence, social support as a facilitator). We also standardised quantitative reporting to odds ratios with 95% confidence intervals ( page 15, lines 344-358).
Comment:
- Consider visualizing the positioning of barriers and facilitators on a CoC framework map. Just as possibilities.
Response: Thank you for this suggestion. While we appreciate the idea of mapping barriers and facilitators onto a CoC framework, we feel that the detailed tables (table 2 and 3, at pages 8-11, lines: 174-179)) already offer a clear and structured presentation of these elements. Adding a visual map may risk cluttering the framework and detracting from interpretability. We hope the current format sufficiently supports clarity and accessibility for readers.
Comment:
- As most of the included studies are from Australia, the UK and the US, it would be useful to consider how national healthcare systems and policy contexts may affect access barriers and facilitators. This is particularly important for generalization to global CALD populations.
Response: Many thanks for highlighting the important context. The Discussion was expanded to explicitly address how health-system and policy contexts in Australia, the United Kingdom, and the United States shape access barriers and facilitators for people with T2D from CALD backgrounds. Specifically, universal coverage, primary care gatekeeping and registration, interpreter provision, cultural competence, out-of-pocket costs, insurance and immigration-linked eligibility, safety-net services, community health worker programmes, care coordination and navigation, and transport or medication subsidies were detailed as mechanisms that hinder or enable access at different points of the care continuum ( page 17, lines: 424-443).
Comment:
- Develop the role of the facilitator. Facilitators are underrepresented compared to barriers and deserve further analysis.
Response: We agree and have developed the facilitators’ role across the continuum of care. We expanded the result and discussion section to explain how facilitators operate, at which stages they act, and how they interact with common barriers. Specifically, we synthesise facilitators at individual (self-efficacy, health literacy), interpersonal (family and peer support, reminders), and system levels (funded interpreter services, culturally competent care, insurance support), and indicate their practical implications for screening/diagnosis, appointment adherence, treatment initiation, and follow-up.
Comment:
- Strengthen the conclusion The conclusion could be more action oriented. Consider summarizing key recommendations for policy, service delivery, and future research. This will be very helpful for policymakers.
Response: The Conclusion was revised to be action oriented by adding explicit, concise recommendations across three domains (policy, service delivery, and future research) ( Page 17, lines: 454-462).
Comment:
- Some parts of the Discussion and Results repeat earlier information. Consider tightening them up to improve clarity and focus. This will allow more room for the discussion I recommended earlier.
Response: Thank you for your feedback. In response to your comment, we have reviewed the Discussion and Results sections to reduce repetition and enhance clarity. This has allowed us to further elaborate on the discussion as recommended.
Comment:
- Have you used any AI tool to improve the grammar and style of the text. I also suggest a careful review of the language, as some phrasing appears overly like source texts, possibly triggering plagiarism detection. Consider paraphrasing for clarity and originality.
Response: Thank you for this observation. In addition to Grammarly and QuillBot, limited language editing was supported by a large language model–based assistant (OpenAI ChatGPT) to improve phrasing and clarity. We have mentioned it in the acknowledgement section of the revised manuscript (page 20, lines: 482-483.
Comment:
- Try to emphasize the results more in the abstract.
Response: Thank you for your suggestions regarding abstract. As you have advised, we have further elaborated the results in the abstract.

Reviewer 2 Report
Comments and Suggestions for Authors
Hello-Thank you for the opportunity to review your article on the barriers and facilitators of CALD populations. Your article will lend nicely to the current publications on this topic. The article was well organized and clearly described the challenges experienced by these populations. The CoC framework fits nicely with your review. Some suggestions worth noting in the article edits include the introduction section, references, supplementary information and choice of population.
In the background you mentioned that the articles were pulled from the date of the database inception. Did you mean each database or a particular one? Could you clarify this statement or list the actual year(s)? Also could you include additional information on the diagnosis? Some of your references are older than 5 years. Are they considered as sentinel articles? If they are not significant to your discussion then you should consider removing them.
You included HSPs as a population in this review but I suggest that you focus on either patients or providers for a better picture of one group. Providers could get added to another article in the future. Also include a breakdown of the patient demographics for the reader to understand which patient populations that you discuss throughout the article.
The organization of the article could improve as some topics are repeated throughout each section. For example, financial constraints and transportation were discussed in several sections. I suggest organizing the barriers into different subheadings to improve the flow of the article as much as possible to reduce reading the same barrier. I agree to use intersectionality in a future article to understand the challenges faced by women at their practitioner’s office.
Overall, this is a great article to continue to understand the experiences of populations with T2D to improve their care. I hope that a reader will take your suggestions in the Discussion area to build on this knowledge.
Author Response
Reviewer 2
Comment:
Hello-Thank you for the opportunity to review your article on the barriers and facilitators of CALD populations. Your article will lend nicely to the current publications on this topic. The article was well organized and clearly described the challenges experienced by these populations. The CoC framework fits nicely with your review. Some suggestions worth noting in the article edits include the introduction section, references, supplementary information and choice of population.
In the background you mentioned that the articles were pulled from the date of the database inception. Did you mean each database or a particular one? Could you clarify this statement or list the actual year(s)? Also could you include additional information on the diagnosis? Some of your references are older than 5 years. Are they considered as sentinel articles? If they are not significant to your discussion, then you should consider removing them.
Response: Thank you for your insightful comments. As you have advised, we have clarified the statement as ‘Searches were conducted from their inception to November 2023 in five databases: Scopus, PubMed, Web of Science, CINAHL and PsycINFO’ manuscript.” (page 3, paragraph 3, lines: 106-107)
As you have advised, we have included additional information on the diagnosis in the revised manuscript (Page-12, paragraph-3, lines: 196-202). Some older references were retained because they offered foundational insights or were identified through our systematic search strategy, including review articles. We have ensured that all cited sources are relevant to the scope and interpretation of our findings.
Comment:
You included HSPs as a population in this review, but I suggest that you focus on either patients or providers for a better picture of one group. Providers could get added to another article in the future. Also include a breakdown of the patient demographics for the reader to understand which patient populations that you discuss throughout the article.
Response: Thank you for your thoughtful feedback. We acknowledge the value of focusing on either patients or providers separately. However, our review offers a unique contribution by capturing perspectives from both patients and healthcare providers, thereby providing a more holistic understanding of the issues explored.
Regarding participant demographics, we have included detailed characteristics of both patients and healthcare providers as reported in the included articles to ensure contextual clarity (Table 1, page 6-8).
Comment:
The organization of the article could improve as some topics are repeated throughout each section. For example, financial constraints and transportation were discussed in several sections. I suggest organizing the barriers into different subheadings to improve the flow of the article as much as possible to reduce reading the same barrier. I agree to use intersectionality in a future article to understand the challenges faced by women at their practitioner’s office.
Response: We appreciate your observation and agree that some barriers, such as financial constraints and transportation, appear in multiple stages of the continuum of care. This repetition reflects the overlap of barriers across different stages of the continuum of care rather than a lack of coherence. We have found that the recurrence of certain barriers underscores their significance.
Comment:
Overall, this is a great article to continue to understand the experiences of populations with T2D to improve their care. I hope that a reader will take your suggestions in the Discussion area to build on this knowledge
Response: Thanks for your complement. We agree with you that knowledge generated from this systematic review will likely contribute to improving care for people with T2D from CALD background
